# "Dependency Bottleneck" in Auto-encoding Architectures: an Empirical Study

**Denny Wu**[*1], **Yixiu Zhao**[*1], **Yao-Hung Hubert Tsai**[*2], **Makoto Yamada**[3], **Ruslan Salakhutdinov**[2]

[1]Computational Biology Department, Carnegie Mellon University
[2]Machine Learning Department, Carnegie Mellon University
[3]RIKEN AIP, JST PRESTO
`yiwu1@andrew.cmu.edu, yixiuz@andrew.cmu.edu,`
`yaohungt@cs.cmu.edu, makoto.yamada@riken.jp, rsalakhu@cs.cmu.edu`

## Abstract

Recent works investigated the generalization properties in deep neural networks (DNNs) by studying the Information Bottleneck in DNNs. However, the measurement of the mutual information (MI) is often inaccurate due to the density estimation. To address this issue, we propose to measure the dependency instead of MI between layers in DNNs. Specifically, we propose to use Hilbert-Schmidt Independence Criterion (HSIC) as the dependency measure, which can measure the dependence of two random variables without estimating probability densities. Moreover, HSIC is a special case of the Squared-loss Mutual Information (SMI). In the experiment, we empirically evaluate the generalization property using HSIC in both the reconstruction and prediction auto-encoding (AE) architectures.

## 1 Introduction

Due to the success of Deep Neural Networks (DNNs), unveiling the generalization properties of DNNs has attracted lots of attention. Recently, Shwartz-Ziv & Tishby (2017) applied mutual information (MI) (Cover & Thomas, 2012) for modeling the training dynamics in DNNs, and two distinct phases are reported. In the first phase, MI between the latent and the output space increases, which correlates with the decrease in the training error. Whereas in the second phase, MI between the input and the latent space decreases, which forces the latent representations to "forget" the input while maintaining the information for the output. It has been suggested that this second phase, known as the "compression" or the "bottleneck", contributes to the generalization performance of the learned latent representation. However, measuring MI between two layers in DNNs is often not easy and can be computationally inefficient. Note that the layers in DNNs refer to high dimensional data, and thus adopting a proper estimator for MI is crucial.

A standard estimation of MI (Cover & Thomas, 2012) requires density estimation of $p(\boldsymbol{x}, \boldsymbol{y})$ and its marginals $p(\boldsymbol{x})$ and $p(\boldsymbol{y})$, and the final estimator is obtained by taking the ratio of the estimated probability densities. However, the approximations may be inaccurate and can lead to a poor MI estimation for high dimensional distributions. Considering this issue, Andrew Michael Saxe (2018) argued that the "compression" in sigmoid neural networks is a result of the binning approximation and the saturation of nonlinearity.

In the paper, instead of measuring MI, we measure the *dependency* between two layers in DNNs. Specifically, we propose to use Hilbert-Schmidt Independence Criterion (HSIC) (Gretton et al., 2005a) as a dependency estimator, which can measure the independenceness between two random variables without density estimations. Moreover, HSIC can be seen as a special case of the squared-loss mutual information (SMI) (Sugiyama & Yamada, 2012). In the experiment, we empirically evaluate the generalization property of the learned latent representations in reconstruction and prediction auto-encoding (AE) architectures. Specifically, we investigate the dependency between different layers for modeling the training dynamics of AEs, examine whether similar "compression" can be observed, and quantitatively compare the latent representations on the recognition task.

---

[*]Equal contribution. Random author ordering.

## 2 RELATED WORKS

### 2.1 INFORMATION BOTTLENECK

Suppose we have a Markov chain $X \to Z \to Y$, where $\boldsymbol{x} \in X$ is the input, $\boldsymbol{z} \in Z$ is the latent representations, and $\boldsymbol{y} \in Y$ is the output, the information bottleneck (IB) (Tishby et al., 2000) can be written as the following optimization problem:

$$\min_{p(\boldsymbol{z}|\boldsymbol{x}), p(\boldsymbol{y}|\boldsymbol{z})} I(X, Z) - \beta I(Z, Y) \tag{1}$$

with $I(\cdot, \cdot)$ representing the mutual information (MI) (Cover & Thomas, 2012). It has been argued that minimizing Eq. (1) corresponds to the "compression" of the input in the latent space and relates to the generalization performance of the model Shwartz-Ziv & Tishby (2017).

### 2.2 AUTO-ENCODING STRUCTURES FOR RECONSTRUCTION AND PREDICTION

An Auto-Encoder (AE) (Bengio & LeCun, 2007) consists of an encoder network $\boldsymbol{f}(\cdot)$ and a decoder network $\boldsymbol{f}'(\cdot)$. The encoder transforms the input into a low dimensional representation, and the decoder recovers the input signal from the latent representation. The training objective can be written as:

$$\min_{\boldsymbol{f}, \boldsymbol{f}'} \quad \frac{1}{n} \sum_{i=1}^{n} L(\boldsymbol{x}_i, \boldsymbol{f}'(\boldsymbol{f}(\boldsymbol{x}_i))) + \beta \phi(\boldsymbol{f}, \boldsymbol{f}'), \tag{2}$$

where $L(\cdot, \cdot)$ represents the reconstruction loss, and $\phi$ represents additional regularization.

It is worth noting that the formation of information bottleneck may not apply in the original AE setting, since the difference between input $X$ and output $Y$ are trained to be minimized. However, if a video stream is used as input, then an AE can be trained to reconstruct $\boldsymbol{x}_i$ (current frame), or to predict $\boldsymbol{x}_{i+n}$ with $n \geq 1$ (future frames). It has been shown that the latent representation of LSTMs trained for both reconstruction and prediction in sequence data yields higher higher classification accuracy than that for reconstruction only; yet the properties of the latent code has not been interpreted in the context of the IB (Srivastava et al., 2015).

## 3 HILBERT-SCHMIDT INDEPENDENCE CRITERION

The Hilbert-Schmidt Independence Criterion (HSIC) (Gretton et al., 2008) is a kernel-based independence measure defined as the squared HS-norm of the cross-covariance operator between two Reproducing Kernel Hilbert Spaces (RKHS). In this paper, we use a normalized empirical estimate of HSIC:

$$\text{HSIC}_{norm}(\text{X,Y}) = \frac{\text{tr}\left(\boldsymbol{KHLH}\right)}{\|\boldsymbol{HKH}\|_F \|\boldsymbol{HLH}\|_F}, \tag{3}$$

where $\boldsymbol{H} = \boldsymbol{I} - \frac{1}{n}\boldsymbol{1}\boldsymbol{1}^\top$, $\boldsymbol{K} \in \mathbb{R}^{n \times n}$ is the Gram matrix of X with $\boldsymbol{K}_{ij} = k(\boldsymbol{x}_i, \boldsymbol{x}_j)$, and $\boldsymbol{L} \in \mathbb{R}^{n \times n}$ is the Gram matrix of Y with $\boldsymbol{L}_{ij} = l(\boldsymbol{y}_i, \boldsymbol{y}_j)$. It is clear that $\text{HSIC}_{norm} = [0\ 1]$. This estimator can be computed in $O(n^2)$ which is computationally efficient whereas the kernel mutual information (KMI) has complexity of $O(n^3)$(Gretton et al., 2005b).

### 3.1 SQUARED MUTUAL INFORMATION AND HSIC

The squared mutual information (SMI) (Suzuki et al., 2009) between two random variables can be written as

$$\text{SMI}(X, Y) = \iint \left(\frac{p(\boldsymbol{x}, \boldsymbol{y})}{p(\boldsymbol{x})p(\boldsymbol{y})} - 1\right)^2 p(\boldsymbol{x})p(\boldsymbol{y})\mathrm{d}\boldsymbol{x}\mathrm{d}\boldsymbol{y}. \tag{4}$$

This is equivalent to changing the $KL$-divergence in the original MI to the Pearson divergence. In this expression, the quantity $r(x, y) = \frac{p(x,y)}{p(x)p(y)}$ can be approximated via kernel density ratio estimation (Sugiyama & Yamada, 2012), $r_{\boldsymbol{\theta}}(\boldsymbol{x}, \boldsymbol{y}) = \sum_{i=1}^{n} \theta_i k(\boldsymbol{x}, \boldsymbol{x}_i)l(y, y_i)$, where parameters of

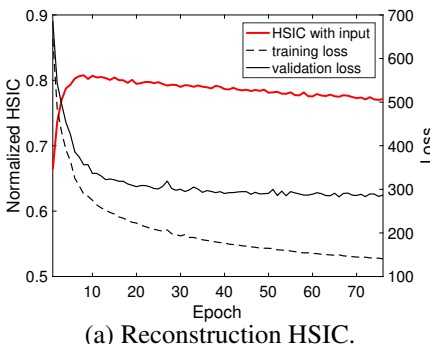
(a) Reconstruction HSIC.

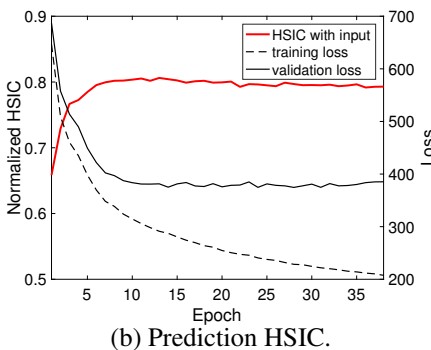
(b) Prediction HSIC.

Figure 1: (a)(b): HSIC between the input frame and its latent representation in AE for reconstruction and prediction. A decrease in HSIC is observed in the training of the reconstructive AE, whereas for the predictive model no obvious decrease in HSIC is observed prior to overfitting.

Table 1: Action Recognition Accuracy Based on one Random Frame

| Latent Representation | Accuracy |
| --- | --- |
| Reconstructive | 0.14 |
| Predictive | 0.29 |

$\widehat{\boldsymbol{\theta}}$ can be learned to minimize the squared-error. In the case of SMI, the optimal $\widehat{\boldsymbol{\theta}}$ can be calculated in closed form, but the solution requires a matrix inverse. The estimator of SMI is given by

$$\widehat{\mathrm{SMI}}(X, Y) = \frac{1}{n} \sum_{i,j=1}^{n} \widehat{\boldsymbol{\theta}}_i k(\boldsymbol{x}_i, \boldsymbol{x}_j) L(\boldsymbol{y}_i, \boldsymbol{y}_j) - 1. \tag{5}$$

Note that if $k(\boldsymbol{x}, \boldsymbol{x}')$ is a centralized kernel matrix and entries in $\widehat{\boldsymbol{\theta}}$ are approximated by $1/n$, then we have $\widehat{\mathrm{SMI}}(X, Y) = \widehat{\mathrm{HSIC}}(X, Y) - 1$. Moreover, we have $\widehat{\mathrm{SMI}}(X, Y) = \widehat{\mathrm{HSIC}}_{norm}(X, Y) - 1$ with $\widehat{\boldsymbol{\theta}} = 1/(n\|\boldsymbol{HKH}\|_F\|\boldsymbol{HLH}\|_F)$. Therefore, HSIC can be interpreted as a special case of SMI (without the optimization of $\widehat{\boldsymbol{\theta}}$ for density-ratio estimation). In experiments on vanilla AEs we indeed found that the trend in HSIC and SMI are similar.

## 4 EXPERIMENTS

AEs for reconstructive and predictive tasks are trained on the UCF-50 dataset (Reddy & Shah, 2013). In both cases a convolution-deconvolution-type architecture is trained. The encoder consists of an encoder and decoder with three convolution layers, and the latent space has 512 hidden units. To speed up training, 12500 frames are chosen from the UCF-50 dataset and each frame is downscaled to 64x64. In the prediction task, the network is trained to predict the next frame after 0.2s. We trained both AEs with Adam (Kingma & Ba, 2014) until the model overfits on the training set.

Contrary to our expectation, a decrease in HSIC is observed in the AE trained for reconstruction, but no significant drop in HSIC is observed in the prediction model before early-stopping (see Fig. 4). To verify the possible connection between this observed drop in dependency and the usefulness of the learned representation, a simple multilayer perceptron (MLP) is trained on the latent representation to perform recognition based on one given frame. We found that the accuracy achieved from the predictive representation is higher than that from the reconstructive representation (see Tbl. 1). We therefore speculate that the drop in HSIC between the input frame and the latent representation might indicate a less useful representation (in classification), even though the reconstruction loss continued to decrease. However, the cause of this drop in dependency, and its apparent absence in the training of AEs for prediction, remains unknown, and the universality of this trend would be interesting future work.

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
