# OpenReview forum: ""Dependency Bottleneck" in Auto-encoding Architectures: an Empirical Study"
_ICLR.cc/2018/Workshop — Reject_

### Official Review · AnonReviewer3 · 2018-03-09
**Interesting idea for estimation of dependency in neural networks.**

**Rating:** 7
**Confidence:** 4

**Review:**

The authors investigate dependence between layers of a deep neural network, and in particular propose to use a normalised Hilbert-Schmidt criterion to measure dependency instead of Mutual Information. They perform experiments using an autoencoder and show the HSIC between the input and the representation, with the results suggesting different behaviours for reconstruction and prediction. The results are still very preliminary but this is a step in an interesting direction.

Questions and comments:
Was the type of kernel used in the experiments Gaussian?
It would be interesting to show results obtained with SMI, especially since the authors seem to have already performed some experiments with SMI.

On formulation:
In the abstract the sentence “we propose to measure dependency instead of MI” is confusing since MI is also a measure of dependency.

---

### Official Review · AnonReviewer1 · 2018-03-10
**Nice motivation, experiments fall short**

**Rating:** 5
**Confidence:** 3

**Review:**

This paper is motivated by recent work which argues that the generalization performance of a DNN can perhaps be understood in terms of its "information bottleneck."  In particular, the recent work of Shwartz-Ziv and Tishby (2017) argues that DNN training has two phases: In the first, the mutual information (MI) between the latent representations and the output increases.  In the second, the mutual information between the input and the latent representations decreases.  Because the MI is difficult to estimate accurately, the authors of this paper propose to measure the Hilbert-Schmidt Independence Criterion ("HSIC") instead.  They measure the HSIC (which is related to the squared mutual information) between the input and the latent representations of two different autoencoders: A reconstruction autoencoder (which reconstructs x_t from x_t), and a prediction autoencoder (which predicts x_{t + n} given x_t).  They observe that for the reconstruction autoencoder, the HSIC begins to consistently + gradually drop after an initial increase at the beginning of training (as predicted by the Shwartz-Ziv and Tishby (2017) paper).  For the prediction autoencoder, on the other hand, no noticeable drop occurs.  The authors were surprised by this difference behavior, and currently offer no explanation, leaving that for future work.

The primary contribution of this paper appears to be its proposal to use HSIC instead of MI to measure the dependence between different layers of a neural net.  This is an interesting proposition.  Unfortunately, the experimental section of this work is quite limited, only considering two models.  Also, the authors were unable to explain their primary observation.  Lastly, I did not understand why they only measured the HSIC between in input and the latent representations, and not between the latent representations and the output.  In particular, why didn't they compute a new version of the "Information Bottleneck" which replaces the MI terms with HSIC terms, and explore whether this metric correlates with generalization performance?

Pros
-- Nice idea of measuring HSIC instead of of MI.
-- Experimental observations are interesting.

Cons
-- Paper doesn't really explain the relationship between HSIC and MI.  What are the important differences between these?  Why should we be able to replace MI with HSIC?
-- Experimental results are quite limited (only two models).  Why are autoencoders the only models considered?
-- Experimental results aren't particularly well understood, and aren't explained very clearly.
-- Paper doesn't perform analysis of correlation between information bottleneck (estimated with HSIC) and generalization performance.
-- The section about approximating SMI seems quite tangential to the work in the paper (also, shouldn't the summand in equation 5 be squared?).

---

### Official Review · AnonReviewer2 · 2018-03-10
**A well-written paper but the empirical study is weak**

**Rating:** 5
**Confidence:** 3

**Review:**

The paper is clearly written and the authors do a good job quickly introducing the pre-requisite concepts (information bottleneck, HS independence criterion, ...). Unfortunately the empirical study itself is quite lightweight. Hopefully the paper can spark some interesting discussions though.

---

### Decision · Program_Chairs · 2018-03-20
**ICLR 2018 Workshop Acceptance Decision**

**Decision:**

Reject

**Comment:**

Based on the reviews, this paper has not been accepted for presentation at the ICLR workshop. However, the conversation and updates can continue to appear here on OpenReview.